# Strategies for the Biofunctionalization of Straining Flow Spinning Regenerated *Bombyx mori* Fibers

**DOI:** 10.3390/molecules27134146

**Published:** 2022-06-28

**Authors:** Paloma Lozano-Picazo, Cristina Castro-Domínguez, Augusto Luis Bruno, Alejandro Baeza, Adelia S. Martínez, Patricia A. López, Ángela Castro, Yassmin Lakhal, Elena Montero, Luis Colchero, Daniel González-Nieto, Francisco Javier Rojo, Fivos Panetsos, Milagros Ramos, Rafael Daza, Alfonso M. Gañán-Calvo, Manuel Elices, Gustavo Víctor Guinea, José Pérez-Rigueiro

**Affiliations:** 1Centro de Tecnología Biomédica, Universidad Politécnica de Madrid, Pozuelo de Alarcón, 28223 Madrid, Spain; paloma.lozano@ctb.upm.es (P.L.-P.); castrocristina24@hotmail.com (C.C.-D.); augusto.bruno@ctb.upm.es (A.L.B.); adeliasm099@gmail.com (A.S.M.); patricia.lopra@gmail.com (P.A.L.); angela.castro@imdea.org (Á.C.); y.lakhal@alumnos.upm.es (Y.L.); elena.monterob@alumnos.upm.es (E.M.); luis.colchero@ctb.upm.es (L.C.); daniel.gonzalez@ctb.upm.es (D.G.-N.); francisco.rojo@ctb.upm.es (F.J.R.); milagros.ramos@ctb.upm.es (M.R.); rafael.daza@upm.es (R.D.); gustavovictor.guinea@ctb.upm.es (G.V.G.); 2Departamento de Ciencia de Materiales, ETSI Caminos, Canales y Puertos, Universidad Politécnica de Madrid, 28040 Madrid, Spain; m.elices@upm.es; 3Neuro-Computing and Neuro-Robotics Research Group, Complutense University of Madrid, 28037 Madrid, Spain; fivos@ucm.es; 4Departamento de Materiales y Producción Aeroespacial, ETSI Aeronautical and Space Engineering, Universidad Politécnica de Madrid, 28040 Madrid, Spain; alejandro.baeza@upm.es; 5Departamento de Tecnología Fotónica y Bioingeniería, ETSI Telecomunicaciones, Universidad Politécnica de Madrid, 28040 Madrid, Spain; 6Biomedical Research Networking Center in Bioengineering, Biomaterials and Nanomedicine (CIBER-BBN), 28037 Madrid, Spain; 7Instituto de Investigación Sanitaria del Hospital Clínico San Carlos (IdISSC), Calle Prof. Martín Lagos s/n, 28040 Madrid, Spain; 8Departamento de Ingeniería Aeroespacial y Mecánica de Fluidos, ETSI, Universidad de Sevilla, Camino de los Descubrimientos, 41092 Sevilla, Spain; amgc@us.es; 9Laboratory for Energy and Environmental Sustainability, Universidad de Sevilla, 41092 Sevilla, Spain

**Keywords:** silk, functionalization, fluorophore, streptavidin, biotin, click chemistry

## Abstract

High-performance regenerated silkworm (*Bombyx mori*) silk fibers can be produced efficiently through the straining flow spinning (SFS) technique. In addition to an enhanced biocompatibility that results from the removal of contaminants during the processing of the material, regenerated silk fibers may be functionalized conveniently by using a range of different strategies. In this work, the possibility of implementing various functionalization techniques is explored, including the production of fluorescent fibers that may be tracked when implanted, the combination of the fibers with enzymes to yield fibers with catalytic properties, and the functionalization of the fibers with cell-adhesion motifs to modulate the adherence of different cell lineages to the material. When considered globally, all these techniques are a strong indication not only of the high versatility offered by the functionalization of regenerated fibers in terms of the different chemistries that can be employed, but also on the wide range of applications that can be covered with these functionalized fibers.

## 1. Introduction

Silk and silk-based materials seem to be ideally suited for biomedical applications. In effect, silk proteins show excellent biocompatibility and, in particular, are not recognized by the immune system [1]. Additionally, silk can be processed in various formats, such as hydrogels, nanoparticles and, most importantly, fibers [2], since the formation of the material from a protein solution depends on the creation of a nanocrystalline phase [3] and does not require the formation of new covalent bonds.

It might seem awkward that a natural material that appears as high-performance fibers spun by either silkworms or spiders [4,5] needs to be dissolved and reconstituted to yield regenerated silk fibers. In this regard, it is usually found that the presence of sericin in natural silkworm silk fibers may be correlated with a negative response of the organism to the implant [6], in spite of the successful usage of sericin in various biomedical applications [7]. A possible explanation for this intriguing behavior may be proposed by assuming that the presence of sericin on the fibers is an indication of a poor purification of the silk fibers. Thus, the contaminants present in the natural material subjected to an incomplete cleaning process, and not sericin itself, would be responsible for the pro-inflammatory effect of the material in the organism. In this context, the dissolution of the raw material and its subsequent spinning ensures the optimum biocompatibility of the regenerated fibers, although it implies developing processing techniques that allow the recovery of the high mechanical performance of the natural material [8].

In addition to an enhanced biocompatibility, silk also offers the advantage of being compatible with several functionalization processes intended to modulate the response of the organism to the biomaterial [9,10,11,12]. The implantation of a material in the body unleashes several negative reactions that may end in the failure of the prostheses [13,14] and are explained in the framework of the “Paradigm of Biocompatibility” [15]. The Paradigm of Biocompatibility states that the reaction of the organism to the implant depends on the recognition by different cell lineages of the biomolecules—mostly proteins—adsorbed on the surface of the material. An immediate consequence of the Paradigm of Biocompatibility is that the response to the material may be tailored by controlling its surface.

The modification of the surface, a process known as functionalization, may proceed through the adsorption of molecules of interest on the material [16]. However, these functionalization methods are not sufficiently robust, since the fate of the molecules in the organism upon implantation of the biomaterial cannot be established easily. Consequently, there is a tendency to prefer functionalization methods that rely on the covalent binding to the material [17]. Unfortunately, the covalent functionalization of the material requires the presence of reactive groups on the surface, usually carboxyls, sulfhydryls or amines. Thus, the functionalization of many common biomaterials, such as polylactic (PLA) and polyglycolic (PLG) acids is considered to be a major challenge, since the functionalization of the hydroxyl groups present in these artificial polymers cannot occur under physiological conditions [18].

Although it is possible to functionalize silk proteins through the hydroxyl-containing amino acids (serines and tyrosines) [19], these processes usually require employing harsh conditions, which hampers their application in combination with biological molecules of interest, such as enzymes. However, silk fibroins present in their sequences a small amount of amino acids that contain reactive groups, such as the amines in the lateral chains of lysine residues, which may react under much milder conditions. Although lysines represent just 0.81% of the molar composition of the heavy chain of *Bombyx mori* fibroin [20], they may be used as anchorage sites for the functionalization of silk fibers. However, the functionalization of natural silk fibers is considered to be a difficult procedure [21], partly due to the reduced number of reactive groups, but also probably due to the high crystallinity of the material [22], which induces a more compact packing of the protein chains. In this context, it is likely that regenerated silk fibers may be functionalized more efficiently than their natural counterparts due to their lower crystallinity [23] and, possibly, due to the lower molecular weight of the processed fibroin [24], that leads to an increase in the number of amine groups at the N-termini of the proteins when considered per unit mass of the fiber.

Following this rationale, this work presents the development of four different strategies for the functionalization of regenerated silkworm (*Bombyx mori*) silk fibers spun through the straining flow spinning process (SFS) [25], leading to the production of high-performance fibers. The functionalization processes are presented sequentially by first indicating the experimental details of the procedure and, subsequently, by showing the results obtained with each strategy. In addition to introducing each individual method, the common procedures used for the preparation of the protein solutions (dope) and for the spinning of these solutions into high-performance fibers are described in the Section 3.

## 2. Results and Discussion

### 2.1. Functionalization with a Fluorophore

Among the low-molecular-weight organic molecules or biomolecules that may be used to functionalize a biomaterial, fluorophores are commonly used to track the performance of the implant, such as when analyzing the real-time degradation of implanted hydrogels and the response exhibited by the surrounding tissues [26,27]. Although several biological structures have an inherent autofluorescence that arises from endogenous fluorophores, such as pyridinic (NADPH) and flavin coenzymes, most commercial exogenous fluorophores, frequently used as cell markers, have stronger fluorescent signals that overcome autofluorescence interference. The stabilization of the fluorophore on the biomaterial usually relies on weak interactions as occurs, for example, when the fluorescent molecule is simply added to the solution employed to create a hydrogel or when the biomaterial is immersed in a solution that contains the fluorophore [28,29]. However, the absence of strong covalent bonds between the molecule and the biomaterial implies several drawbacks, such as the possible loss of the fluorophore during processing or upon implantation, or the possible aggregation of the molecules that may lead to quenching and the corresponding reduction on the intensity of fluorescence [30,31]. In this regard, the functionalization of regenerated silk fibers offers the possibility of producing fibers whose fate can be easily tracked and, at the same time, constitutes a first step in the development of protocols and techniques required to create more complex functionalization systems.

Fluorescein 5(6)-isothiocyanate (FITC) is a common fluorophore that may bind covalently to amines through the isothiocyanate group. Two different protocols were developed to functionalize the regenerated fibers with FITC, that differed in the step chosen to covalently bind the molecule to the fibroin: (A) proteins are functionalized in solution and the fibers are subsequently spun from that dope, or (B) regenerated fibers are functionalized directly by immersion in a FITC solution. Both processes are described in detail below.

Protocol A starts with a 4% fibroin solution in 50 mM Tris Base buffer (C_4_H_11_NO_3_, Fisher Scientific) at pH 8. 1 mg of FITC was added to the solution and allowed to react for 20 min in mild agitation and darkness. The solution was subsequently dialyzed for 24 h against 50 mM Tris Base buffer at pH 8, using SnakeSkin^TM^ Dialysis Tubing (3500 MWCO, Thermo Fisher Scientific, San Jose, CA, USA) to remove the excess of FITC. The concentration of the functionalized fibroin dope was increased up to a value of 15% through a final step of reverse dialysis against a 50 mM Tris base buffer solution at pH 8 of PEG 8000 (Polyethylene glycol, Fisher Bioreagents; PEG concentration 10% *w*/*v*) containing 1 M of calcium chloride dihydrate (CaCl_2_‧2H_2_O, Acros Organics, Morris Plains, NJ, USA). The dope formed from the functionalized fibroin was used to spin the SFS regenerated fibers as described below.

Protocol B was developed to functionalize the SFS fibers with the fluorophore. In this case, 50 mg of SFS regenerated fibers were immersed in a solution of 0.5 mg/mL FITC in PBS at pH 7.4 for 20 min in mild agitation and darkness. The fibers were removed from the solution, and washed three times for five minutes each in PBS. Lastly, the fibers were rinsed with distilled water and allowed to dry overnight. Protocol B was also used to functionalized natural silk and observe to yield fluorescent fibers. However, it was not confirmed whether the fluorophore was exclusively bound to the fibroin proteins or, additionally, to the other biomolecules that constitute the fiber, such as proteins of the sericin family.

The results of both functionalization protocols are presented in Figure 1, that compares the fluorescence observed on a yarn of SFS fibers functionalized with protocol A, a yarn of SFS fibers functionalized with protocol B, and a yarn of control (non-functionalized) SFS fibers.

To get a quantitative measurement of the efficiency of both protocols, 40 mg of fibers functionalized with the protocols A and B were dissolved in LiBr following the same procedure described below for the solution of the raw silkworm cocoons presented in the Section 3, and dialyzed against distilled water. Absorbance of the fibroin solution obtained from each type of fibers was measured at 492 nm in a Halo RB-10 spectrophotometer and the fluorescein present in each case was quantified through a standard FITC curve at this wavelength prepared with a series of solutions of known concentration. Finally, the absorbance was converted into concentration of fluorescein and normalized to the fibroin content. The results are shown in Table 1.

Probably the most significant result obtained from the analysis of the functionalization of regenerated fibers with FITC is the verification that the amine groups present in the protein, either appearing in the N-termini or in the lateral groups of basic amino acids, can be effectively used to bind biomolecules covalently. The results above establish that the functionalization may proceed alternatively through the initial modification of the protein in solution and subsequent straining flow spinning of the functionalized fibroin, or through the direct functionalization of previously spun SFS fibers. The former (functionalization of the protein in solution) leads to an increase in the proportion of covalently bound molecules but, it can be hypothesized that a significant proportion of the functionalizing molecules may be buried inside the fiber. The location of the molecules inside the fiber does not necessarily affect the efficiency of the fluorophore, but it renders these molecules useless if their function requires the interaction with the environment that surrounds the fiber, as will be discussed below. Consequently, the direct functionalization of regenerated fibers with the molecules of interest will be common to the strategies presented in the following sections.

### 2.2. Functionalization with the System Streptavidin-Biotin

The streptavidin-biotin system provides a functionalization strategy compatible with a wide range of materials and applications [32]. Although the interaction between biotin and streptavidin results from a combination of hydrogen bonds, London forces and conformational changes, its dissociation constant with a value of K_d_ ≈ 10^−15^ M is comparable to those characteristic of covalent bonds [33]. In addition, the streptavidin-biotin complex remains stable within an ample range of pH values and temperatures. Since the streptavidin molecule is a tetramer that contains four independent binding sites to biotin, it is possible to use one of these sites to bind the molecule to the material, while maintaining enough active sites to allow the interaction of streptavidin with the bioactive molecule of interest.

Following the previous results obtained with the FITC molecules, the functionalization of the regenerated fibers with the streptavidin-biotin system proceeded with the initial binding of biotin to the fibroin proteins through the amine groups and the assessment of the accessibility of the immobilized biotin through its interaction with labelled streptavidin molecules. The functionalization protocol starts with the preparation of a solution of NHS-biotin ((poly(ethylene glycol)-(N-hydroxysuccimide 5-pentanoate)-ether 2-(biotinylamino)ethane; Sigma Aldrich, Saint Louis, MO, USA) in PBS at pH 7.4 by dissolving 3 mg of NHS-biotin in 540 μL of the buffer. The regenerated fiber is immersed in a volume of 135 μL of the solution and incubated for 2 h at 4 °C. This initial step allows the reaction of the N-hydroxysuccinimide (NHS) group with the amines of the fibroin, so that a covalent bond is formed. After being removed from the solution, the fibers were washed three times for five minutes each in PBS and rinsed with distilled water.

To assess the correct biotinylation of the regenerated fibers, streptavidin-FITC molecules (Streptavidin from *Streptomyces avidinii*; Sigma Aldrich, Saint Louis, MO, USA) was employed. An 0.5 mg/mL solution of streptavidin-FITC in PBS at pH 7.4 was prepared and the fiber was incubated in 100 μL of the solution for one hour at 4 °C and in darkness. The samples were washed three times for five minutes each in PBS. The fibers were then observed in a fluorescence microscope as detailed in the Section 3. Non-biotinylated fibers incubated with the streptavidin-FITC solution were used as control and observed under the same conditions. Representative fluorescence micrographs are shown in Figure 2. The fluorescence of these images was quantified with the ImageJ software and the results are summarized in Table 2.

The effective immobilization of streptavidin molecules on the surface of SFS fibers opens the possibility of combining these fibers with a wide range of bioactive molecules that may be purchased (or modified in the laboratory) as biotinylated derivatives. However, the relatively large size of the streptavidin might represent a drawback for some systems, so that it is convenient to develop yet another functionalization strategy that relies on molecules of lower molecules weight.

### 2.3. Functionalization with the Biorthogonal Alkyne-Azide Chemistry

The concept of bioorthogonal (or “click”) chemistry is applied to a set of chemical reactions that lead to the formation of highly selective covalent bonds in a lapse of time not longer than 2 h [34]. Among the various bioorthogonal chemistries available, the cycloaddition between an azide and an alkyne group shows the significant advantage that none of these groups is present in biological systems, so that no interference is induced upon the addition of these moieties to the system [35].

The original azide-alkyne click reaction required the presence of copper ions as catalyzer [36,37], but the presence of these ions was found to exert a toxic effect for the survival of cells [38]. It was subsequently found, however, that the addition of copper ions was not necessary if the alkyne appeared in a stressed conformation, such as that found in a cyclic molecule with eight carbon atoms [39]. In this regard, the development of the azide-stressed alkyne click chemistry offers a neat opportunity to functionalize a wide range of materials under physiological conditions.

Following the previous functionalization strategies, the amine groups present in the silk proteins either at the N-termini or as lateral groups of basic amino acids were chosen as targets to immobilize the components of the click chemistry. The selected crosslinkers were NHS-azide (azidoacetic acid NHS ester) and NHS-alkyne (dibenzocyclooctyne-N-hydroxysuccinimidyl ester), whose chemical formulas are presented in Figure 3.

Although both the azide and alkyne compounds may be bound to the amines present in the fibroin, it is necessary to check if at least one of these compounds is compatible with the bioactive molecule of interest. In this case, lactate dehydrogenase (LDH) was chosen as model molecule to show the possibility of producing regenerated silk fibers with enzymatic activity through the use of the click chemistry.

LDH catalyzes the conversion of pyruvate into lactate following the reaction:Pyruvate + NADH + H^+^ → Lactate + NAD^+^
and is especially suited as model molecule for the analysis of the interaction between materials and biological systems due to the possibility of detecting even less than one nanomole of the protein through an enzymatic procedure [40]. The application of LDH for this type of analysis relies on both the large specific activity of the enzyme, with values in the range of 200 μmol_NADH_/min.mg_prot_ and in the large absorption coefficient of NADH at a wavelength of λ = 340 nm (ε = 6.22 mM^−1^ cm^−1^). Thus, monitoring the variation with time of the NADH concentration, it is possible to quantify the number of LDH molecules present in the system. The details of the experimental procedure may be found elsewhere [40].

LDH purified from bovine heart (Sigma Aldrich, Darmstadt, Germany) was functionalized with either NHS-alkyne or NHS-azide following the protocols below. The functionalization with NHS-alkyne was performed in a Na_2_CO_3_ buffer (0.01 M; pH = 8.5) by adding 17 μL of the stock LDH solution (12.1 mg/mL) to 250 μL of the buffer. The alkyne was dissolved in DMSO to a concentration of 5 mg/mL and 100 μL of this solution was added to the initial solution of LDH in carbonate buffer, and the mixture was incubated overnight under agitation. The solution was then dialyzed at 4 ºC to purify the LDH bound to the alkyne against a buffer of 5% PBS (pH 7.5) in distilled water with a final volume of 200 mL. The dialysis process proceeded overnight. The functionalization of LDH with azide also started by adding 17 μL of the stock LDH solution (12.1 mg/mL) to 250 μL of carbonate buffer. A solution of 1.82 mg/mL of NHS-azide in DMSO was prepared and 12.5 μL of this solution was added to the LDH solution in carbonate buffer. The mixture was incubated overnight under agitation and dialyzed at 4 ºC against 200 mL of a 5% PBS (pH 7.5) in distilled water.

The alkyne- and azide-functionalized LDH were recovered after the dialyses step and its specific activity was determined from the variation of NADH in the solution as described in [40]. The results are presented in Table 3, where the specific activity measured from the non-functionalized LDH stock is also presented.

The results presented in Table 3 indicate that the functionalization with alkyne leads to the loss of the catalytic ability of the enzyme, while no such effect is observed in the azide-functionalized protein. The loss of function upon functionalization with alkyne might be related with the larger size of the NHS-alkyne compared with the azide molecule, so that it may be hypothesized that the larger alkyne group may present some steric hindrance for the correct function of the enzyme.

As a consequence of the previous analysis, the fibers were functionalized with the alkyne group, so that a covalent bond could be created between the fibers and the azide-functionalized LDH. To functionalize the material, 10 mg of fibers were incubated in a solution of 500 μL of DMSO and 4 μL of N,N-Diisopropylethylamine (DIPEA). DIPEA is a base that deprotonizes the amine groups in the lysines, which increases its nucleophilicity and enhances its reactivity with the NHS group. Subsequently, 32 μL of a 5 mg/mL solution of NHS-alkyne were added to the original solution and maintained at room temperature overnight under agitation. After removing the DMSO solution, the fibers were washed three times with distilled water and stored at 4 °C after being used.

Before testing the bond between azide-functionalized LDH and alkyne-functionalized fibers, the reactivity of the alkyne groups bound to the fiber was preliminary assessed using fluorescein-azide. The fibers were washed in 1 mL of DMF (N,N-dimethylformamide) to remove the possible leftovers of DMSO. Subsequently, the fibers were immersed in 500 μL of DMF and 10 μL of a fluorescein-azide solution (1 mg fluorescein-azide/50 μL DMF), and incubated in darkness at room temperature for 2 h under agitation. After completing the incubation, the fibers were washed with 1 mL of DMF three times. The fluorescence of the fibers was measured with a fluorescence microscope DMI3000B Inverted Leica DMIRB equipped with a digital camera Leica DC100 (Leica, Germany) using the following observation parameters: exposure: 8.61 ms; gamma 0.60; gain 1.0X. The results are shown in Figure 4 and the fluorescence of each image was quantified with the ImageJ software.

**Figure 4 molecules-27-04146-f004:**
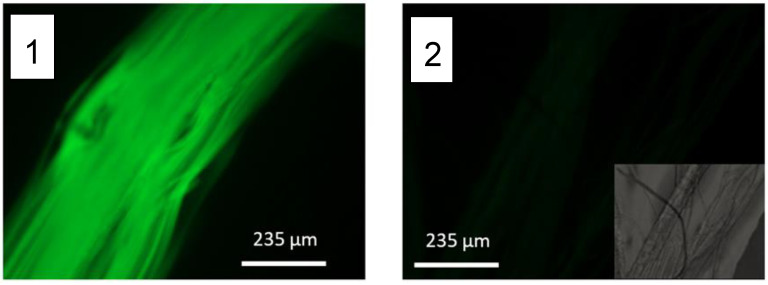
Fluorescence micrographs of (**1**) Yarn of fibers functionalized with alkyne and incubated with fluorescein-azide. (**2**) Yarn of control fibers (non-functionalized with alkyne) incubated with fluorescein-azide. The insert in Figure (**2**) shows the bright field optical micrograph of the fibers, due to the low fluorescence observed in the control fibers. The fluorescence of the micrographs is quantified in Table 4.

As the final step of the functionalization process, 10 mg of alkyne-functionalized fibers were incubated in 600 μL of a 100 μg/mL azide-functionalized LDH solution in PBS (pH 7.5) for two hours at room temperature. To remove the LDH not covalently bound to the material, the fibers were immersed in 40 mL of PBS at 4 °C overnight. The fibers were subsequently retrieved from the solution and introduced in an Eppendorf tube adding 1 mL of PBS (pH 7.5) and maintained under these conditions until measuring their catalytic activity. Fibers non-functionalized with alkyne were subjected to the same protocol and used as control samples.

Both alkyne-functionalized and control fibers after being incubated with azide-LDH were immersed in 600 μL of a 6 mM solution of NADH in PBS (pH 7.5) containing 2.9 mM pyruvate. The variation in the concentration of NADH was measured as described above and used to determine the activity of the enzyme. The results shown in Table 5 indicate that the protocol developed for the functionalization of the fibers based on the click chemistry leads to the production of regenerated fibers with a significant catalytic ability.

### 2.4. Functionalizatin with Cell-Adhesion Motifs

Among the different applications suited to functionalized regenerated silk fibers, the modulation of the behavior of various cell lineages through their interaction with the fibers is likely to represent a promising procedure for the development of new therapies. The functionalization of biomaterials with bioactive proteins and peptides allows modulating the response of the organism to the material and leads to increased rates of survival and proliferation on the implant [41,42]. Following this rationale, a strategy was developed to functionalize the fibers with the RGD peptide, and the performance of the fibers was assessed through cell cultures with mesenchymal stem cells (MSCs). We chose this heterogeneous cell population due to the enormous repercussion of MSCs in the field of regenerative medicine in various clinical conditions [43,44].

The RGD peptide (full sequence: GRGDSP, Sigma Aldrich, Darmstadt, Germany) was immobilized on the fibers using the EDC/NHS crosslinking chemistry (EDC: N-(3-dimethylaminopropyl)-N’-ethylcarbodiimide hydrochloride, Sigma Aldrich; NHS: N-hydroxysuccinimide, Sigma Aldrich), as explained in detail elsewhere [45]. Briefly, silk fibers were incubated with the RGD peptide diluted in MES buffer (0.1 M, pH 5.0–6.0) to a final concentration of 200 μg/mL for one hour at room temperature. Subsequently, the EDC/NHS crosslinkers were added to a final concentration of 0.125 mg/mL and 0.0315 mg/mL, respectively. After an incubation of 4 h, the fibers were washed with distilled water and immersed twice in PBS for 2 h each. To remove any possible leftover of the crosslinking agents, the fibers were immersed for 72 h in 0.1 M MES at 4 °C. Finally, the samples were immersed twice in DMEM for 12 h each.

To check the presence of the peptide on the fibers, the RGD peptide was bound to fluorescein as explained in detail elsewhere [45]. Briefly, the peptide was diluted in PBS to a concentration of 500 μg/mL and FITC was added to a final concentration of 250 μg/mL. Figure 5 compares the fluorescence micrographs obtained from the control samples incubated with the RGD-FITC peptides, but not with the EDC/NHS crosslinkers, and those obtained by adding the crosslinking molecules. The difference in fluorescence between both micrographs indicates that the functionalization of the fibers with the RGD peptide proceeds efficiently through this method by binding the C-termini of the peptide to the amine groups of the fibers.

To assess the behavior of MSCs on the fibers, cells were seeded on dry sterilized fibers functionalized with RGD (the peptides used in this second experiment were not labelled with FITC) in a 24-well plate at a density of 2 × 10^4^ cells/well in 500 mL of basal medium containing DMEM, FBS, L-glutamine, P/S as described elsewhere [46]. After 24 h cells were dyed with calcein AM for 30 min at 37 °C and fluorescent, as well as bright field images, were taken. Representative micrographs are shown in Figure 6, and the number of cells attached on control (non-functionalized fibers) and RGD-functionalized fibers are presented in Table 6. From these results it is apparent that the immobilization of the RGD peptide on the fibers using the EDC/NHS crosslinking chemistry leads to a significant increase in the number of MSC cells attached to the material.

## 3. Materials and Methods

### 3.1. Preparation of the Silkworm Silk (Bombyx mori) Protein Solution

Silkworm silk *(Bombyx mori*) cocoons with a weight of 5 g were cut in four pieces and degummed in 1 l of deionized water containing 2 g of Na_2_CO_3_ for 20 min. The fibers were washed with 500 mL of water for 5 min (process repeated thrice) and allowed to dry overnight. The degummed silk fibers were immersed in a 9.3 M LiBr solution in water at a concentration of 20% (*w*/*v*) and heated at 60 °C for 4 h. The fibroin solution was dialyzed against distilled water at 4 °C to remove the lithium and bromide ions. Dialyses proceeded in six steps, changing the dialysis medium each time. The solution was centrifuged at 5000 rpm for 20 min, while being maintained at 4 °C. The supernatant was retrieved and centrifuged under the same conditions a second time. The concentration of the silk fibroin is approximately 4% (*w*/*v*).

To increase the fibroin concentration, 500 mL solution of a 10% (*w*/*v*) polyethylene glycol (PEG) with a molecular weight of 8000 Da in 50 mM Tris base (pH 8.0) and 1 M CaCl_2_ was prepared. The silk fibroin solution was subjected to reverse dialyses against this PEG solution for 24 h at 4 °C. The fibroin concentration after completing the reverse dialysis step is approximately 16% (*w*/*v*).

### 3.2. Spinning of High-Performance Regenerated Silk Fibers through Straining Flow Spinning

Straining flow spinning (SFS) is a biomimetic process that allows the efficient production of fibers from protein solutions [25]. In parallel with the natural spinning by spiders and worms [47,48], the protein solution in SFS undergoes chemical modifications and is subjected to controlled mechanical stresses that, in combination, lead to the formation of the solid fiber from the dope. The core of the SFS process lies in a coaxial structure formed by a capillary and a nozzle with convergent geometry [49]. The interaction between the dope jet and the focusing fluid in the space created by the capillary and the nozzle, as well as the mechanical stresses exerted on the dope jet through the convergent geometry of the nozzle, control the self-assembly of the proteins and allow obtaining high-performance silk fibers, both from natural fibroins [49] and from genetically engineered proteins [50].

SFS is a highly versatile technique in which the spinning process is controlled by a large number of parameters divided into three groups: geometrical, chemical and hydrodynamical. The geometrical parameters include, among others, the distance between the end of the capillary and the nozzle outlet and the diameter of the nozzle outlet. The chemical parameters comprise the composition of the dope, of the flow focusing and of the coagulating bath, although frequently the composition of the coagulating bath coincides with that of the focusing fluid. Lastly, the hydrodynamical parameters include the flow rates of the dope and focusing fluids, as well as the speed of the take up mandrel. It is common to include an additional post-spinning drawing step to enhance the mechanical properties of the fiber [51].

Regenerated silk fibers were spun through the SFS process, using the fibroin dope prepared as described above. The dope is injected in the silica capillary (inner diameter 150 μm) using a Becton-Dickinson (BD) 1 mL syringe with a Harvard Apparatus 11 Plus syringe pump. The end of the capillary is located at a distance of 2 mm from the nozzle outlet and this outlet has a diameter of 500 μm. The focusing fluid is injected with another Harvard Apparatus 11 Plus pump acting on a BD 50 mL syringe. The composition of the focusing and coagulating bath is acetic acid (1M) in water/ethanol (20/80 *v*/*v*). The flow rate of the dope is Q_d_ = 5 μL/min and the flow rate of the focusing fluid Q_f_ = 0.5 mL/min. The linear speed of the take up mandrel is V_1_= 5 cm/s and the fiber was subjected to a post-spinning step in water [51] at a linear speed of V_2_ = 9 cm/s. A detailed description of the tensile properties obtained through the straining flow spinning process can be found elsewhere [52,53], including the influence of the processing parameters on the mechanical behavior of the material. Figure 7 illustrates the characteristic tensile behavior found in fibers spun with the conditions indicated above, measured as described elsewhere [54]. In particular, it is found that fibers with a work to fracture in the range of W_f_ = 50–60 MJ/m^3^ are produced.

### 3.3. Measurement of Fluorescence

Except where indicated otherwise in the text, fluorescence was measured in a microscope DMI3000B Inverted Leica DMIRB equipped with a digital camera Leica DC100 using the following parameters: exposition time 1 s; Gain 2.1; Gamma 0.83. Images were taken at a wavelength of 520 nm that corresponds to the emission peak of fluorescein.

## 4. Conclusions

The possibility of functionalizing regenerated silk fibers with various alternative strategies opens a new dimension for the application of these materials in the biomedical field, in addition to their excellent biocompatibility and mechanical properties. Our results show that it is feasible to covalently bind an ample variety of biomolecules to the silk fibers through the amine groups present in the proteins, either at the N-termini or as lateral groups of the lysine residues.

Thus, it is possible to produce fluorescent fibers by immobilizing a fluorophore to the fibroin. In addition, the compatibility of the regenerated fibers with the streptavidin-biotin system allows combining the material with the large set of biotinylated molecules. Lastly, the functionalization of the regenerated fibers with enzymes and cell-adhesion motifs creates new opportunities for the application of these materials in various fields of Regenerative Medicine and Tissue Engineering.

The set of strategies presented in this work neither exhausts all the crosslinking chemistries compatible with regenerated silk fibers, nor exploits their full range of applications, but it should be considered as a clear indication of the vast number of possibilities opened by the usage of these biomaterials in the Future.

## Figures and Tables

**Figure 1 molecules-27-04146-f001:**
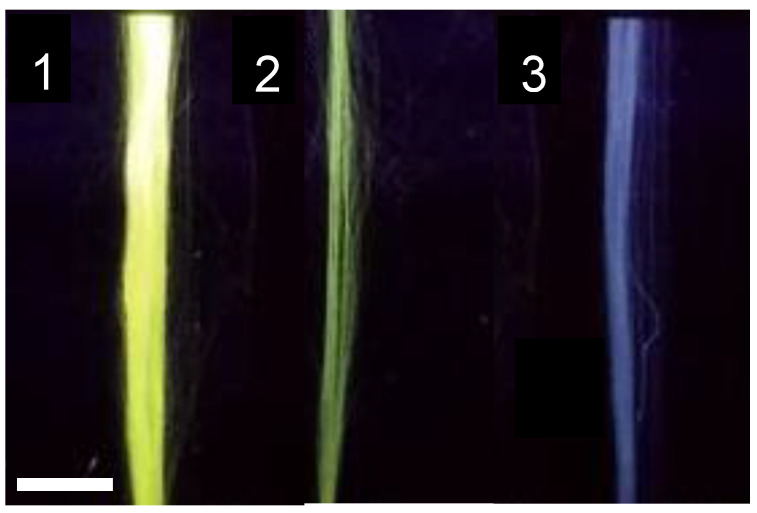
Illustrative fluorescence micrographs of SFS regenerated fibers illuminated with λ = 254 nm UV light (**1**) Yarn of fibers functionalized following Protocol A (Functionalization of the protein in solution and subsequent spinning), (**2**) yarn of fibers functionalized following Protocol B (Immersion of the fibers in an FITC solution), and (**3**) yarn of control (non-functionalized) fibers. The scale bar corresponds to 5 mm.

**Figure 2 molecules-27-04146-f002:**
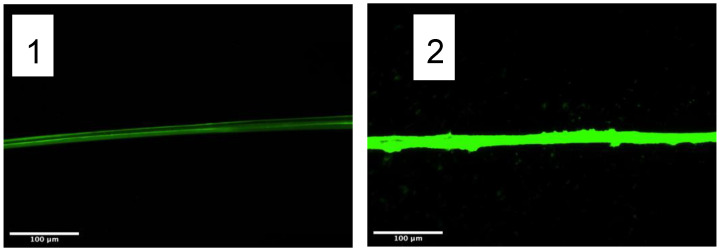
Fluorescence micrographs of SFS regenerated fibers obtained at λ = 520 nm (**1**) control (non-biotinylated fiber) incubated with streptavidin-FTIC. (**2**) Biotin-functionalized SFS fiber incubated with streptavidin-FITC.

**Figure 3 molecules-27-04146-f003:**
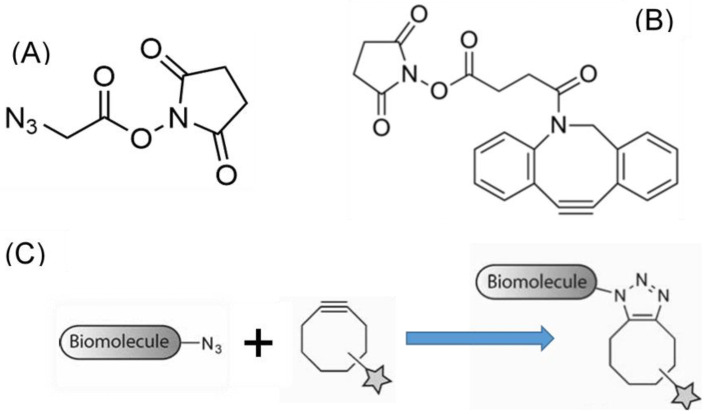
(**A**) Structural formula of the azidoacetic acid NHS ester (NHS-azide) molecule. (**B**) Structural formula of the dibenzocyclooctyne-N-hydroxysuccinimidyl ester (NHS-alkyne), and (**C**) Scheme of the cycloaddition reaction between the azide and stressed alkyne groups.

**Figure 5 molecules-27-04146-f005:**
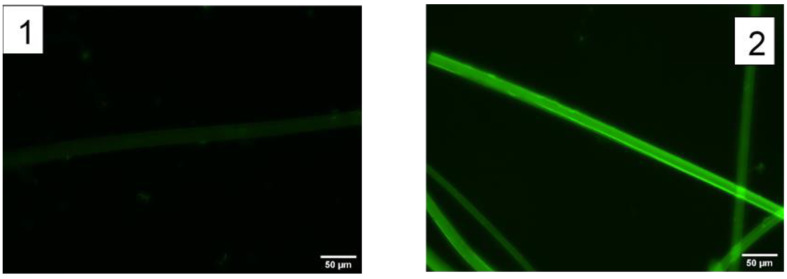
Illustrative fluorescence micrographs of (**1**) a control fiber incubated with fluorescent-RGD peptide in the absence of the EDC/NHS crosslinkers, and (**2**) a fiber incubated with fluorescent-RGD peptide in a solution containing the EDC/NHS crosslinkers.

**Figure 6 molecules-27-04146-f006:**
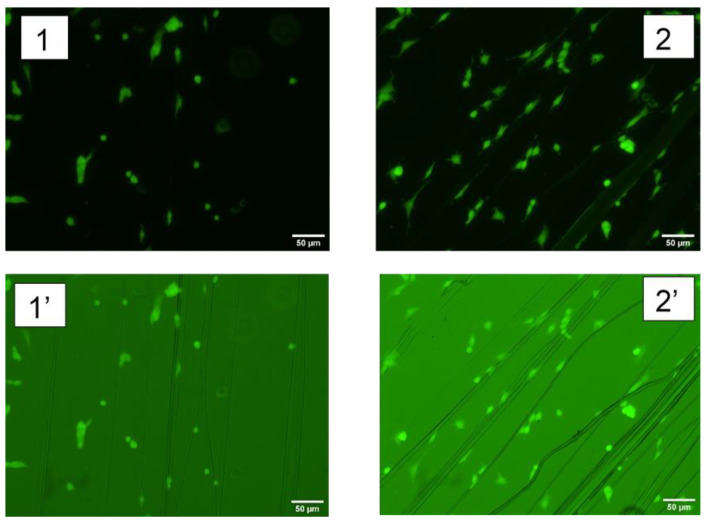
MSC cells cultured in the presence of regenerated silk fibers. Non-functionalized fibers, (**1**) fluorescent micrograph, and (**1′**) bright field micrograph. RGD-functionalized fibers, (**2**) fluorescence micrograph, and (**2′**) bright field micrograph.

**Figure 7 molecules-27-04146-f007:**
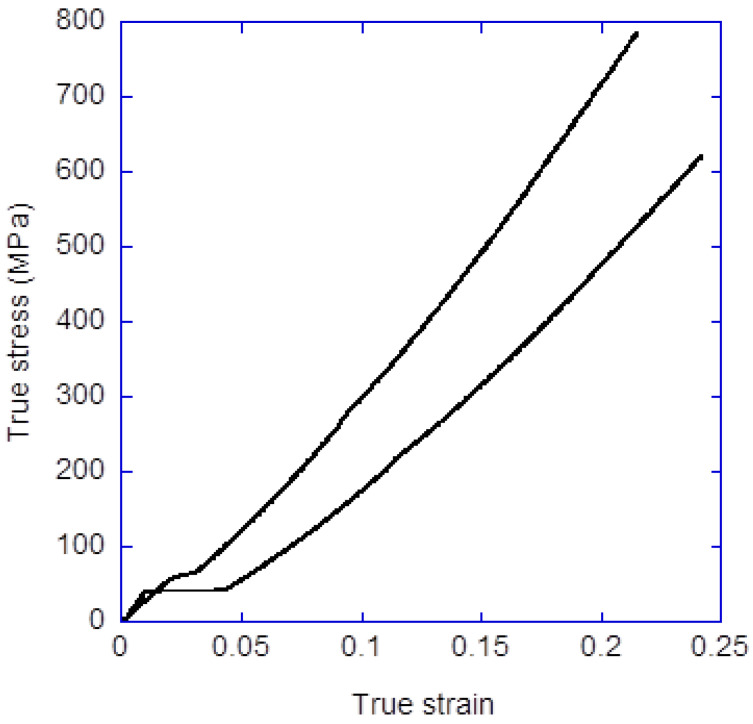
True stress-true strain curves of representative regenerated SFS fibers spun with the processing parameters detailed in the main text. True strain is defined as the force to which the fiber is subjected, divided by the instantaneous cross sectional area, and true strain as the natural logarithm of the instantaneous length, L, divided by the initial length, L_0_, i.e., ε = Ln(L/L_0_).

**Table 1 molecules-27-04146-t001:** Quantitative comparison of the fluorescein molecules covalently bound to fibroin using the functionalization protocols A (Functionalization of the protein in solution and subsequent spinning) or B (Immersion of the spun fibers in an FITC solution). Each value corresponds to the readout of the spectrophotometer obtained from a dissolution of 40 mg of a yarn of fibers functionalized with the corresponding procedure.

Protocol	Fluorescein/Fibroin (μg/mg)
A	0.060
B	0.042

**Table 2 molecules-27-04146-t002:** Quantitative comparison of the fluorescence intensity of the control and biotinylated SFS fibers after being incubated with streptavidin-FITC. Two micrographs were taken from different regions of the fiber and, at least, three measurements of fluorescence intensity were obtained from each micrograph. Values are expressed as mean ± standard error.

SFS Fiber	Fluorescence Intensity (a.u.)
Control (Non-biotinylized)	49 ± 6
Biotin-functionalized	130 ± 20

**Table 3 molecules-27-04146-t003:** Comparison of the specific activities of LDH functionalized with either NHS-alkyne or NHS-azide. The specific activity of the protein before being functionalized is also presented.

Protein	Specific Activity (μmol_NADH_/min.mg_prot_)
LDH (Stock)	220 ± 30
Alkyne-functionalized LDH	1.3 ± 0.9
Azide-functionalized LDH	180 ± 30

**Table 4 molecules-27-04146-t004:** Comparison of the fluorescence observed on fibers functionalized with alkyne and incubated with fluorescein-azide and control (non-functionalized) fibers incubated with fluorescein-azide. Two micrographs were taken from different regions of the fiber and at least three measurements of fluorescence intensity were obtained from each micrograph. Values are expressed as mean ± standard error.

Fiber	Fluorescence Intensity (a.u.)
Alkyne functionalized + fluorescein azide	42 ± 3
Control fiber + fluorescein azide	8 ± 2

**Table 5 molecules-27-04146-t005:** Comparison of the enzymatic activity of alkyne-functionalized fibers and control (non-functionalized) fibers incubated with azide-LDH.

Fiber	pmole_NADH_/min.mg_fiber_
Alkyne-functionalized	420 ± 70
Control	70 ± 50

**Table 6 molecules-27-04146-t006:** Comparison of the number of MSCs attached to the control and RGD-functionalized fibers after 24 h of cell culturing. The cells are counted from micrographs with the same scale as those shown in Figure 6.

Fibers	Number of Cells Attached per Fiber	Number of Cells Attached to Fibers/Total Number of Cells
Control	1.6 ± 0.2	0.27 ± 0.05
RGD-functionalized	2.5 ± 0.3	0.61 ± 0.04

## Data Availability

Data are available upon request to the Corresponding Author.

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
