# Peer review of "Strategies for the Biofunctionalization of Straining Flow Spinning Regenerated Bombyx mori Fibers"

_molecules, 2022, doi:10.3390/molecules27134146_

Round 1

Reviewer 1 Report

The authors presented an interesting study on functionalization of strain flowing spinning regenerated silk fibers. This topic is worthy investigation. However, some major concerns are listed below for the authors to consider:

1. What is SFS? Please define it first before using the abbreviation.

2. From Table 1, the covalently bounded fluorescein in Protocol A is slightly more than that in Protocol B. However, in Fig. 1, the fibers functionalized using Protocol A is much brighter than the fibers functionalized using Protocol B. What is the reason? 

3. For all the functionalizations demonstrated in this study, have the authors do with the natural silk fibers? Is there a big difference between natural fibers and SFS fibers? It is needed to compare two types of fibers.

4. How about the mechanical properties of the functionalized SFS fibers? The tensile testing needs significant improvement to include multiple samples, errors, and statistics.

Author Response

Reviewer #1

We appreciate the overall consideration of our work expressed by Reviewer #1 and have followed his/her indications as follows:

  1. What is SFS?. Please define it first before using the abbreviation.

As indicated by Reviewer #1 the definition of the acronynm in the original version appeared in the Materials and Methods section. The definition is now introduced first in the Abstract and, additionally, in the Introduction (Line 99).

  1. Difference in brightness between the fluorescent samples prepared with Protocol A and B.

Figure 1 is intended to illustrate the difference between the samples prepared with Protocol A and Protocol B with the control samples. In contrast with other figures in the manuscript no quantitative data were obtained from these images. Thus, the observation conditions were slightly modified in order to improve the quality of the micrograph. The illustrative character of the micrographs is indicated explicitly in the Figure Caption 1 of the revised version (Line 171).

  1. Comparison of the functionalization process on natural silk fibers.

Since the paper is focused on the possible biomedical applications of silk, we have concentrated on the functionalization of regenerated silk fibers due to their superior biocompatibility when compared with natural silk fibers. However, and although not included in the original version, we explored the possibility of using Protocol B for producing fluorescent fibers from the natural material. The fibers obtained are fluorescent, but we did not confirm whether the binding between the fluorophore and the material was through the fibroin proteins, since the natural fibers may also contain proteins of the sericin family. This additional information is now included in the revised version (Line 153).

  1. Mechanical properties of the SFS fibers.

As indicated by the Reviewer, the subject of the mechanical properties of the SFS fibers was probably not treated in sufficient extension in the original version. Although a couple of references by our group, in which this information was provided, can found in the initial version, it was not straightforward to retrieve this information. In the revised version we have rewritten the last paragraph of the discussion (Line 468). In addition, we state explicitly that a detailed description of the tensile properties of SFS fibers, and the possibility of modifying these properties by varying the processing parameters, is found in these previous works. In this regard, we have expanded the original list of references with two more references to stress the dependence of processing on the behaviour of the material. We stress that high performance fibers may be produced with the concrete conditions stated explicitly in the manuscript as illustrated in Figure 7.

Reviewer 2 Report

The manuscript entitled Strategies for the biofunctionalization of straining flow spinning regenerated Bombyx mori fibers submitted by the group of Authors represents an interesting research on the development of four different strategies for the functionalization of regenerated silkworm (Bombyx mori) silk fibers spun through the straining flow spinning process, that leads to the production of high performance fibers.

The Introduction part should be enriched with recent references.

Author Response

Reviewer #2

We are grateful for the extremely positive consideration of Reviewer #2 towards our work. Following his/her suggestion we have included the following 5 additional references in the Introduction.

  • Furuzono et al. 1999 J Appl. Polym Sci.,
  • Murphy et al. 2008 Biomaterials,
  • Murphy et al. 2009 J Mater Chem,
  • Chen et al. 2018 Adv Eng. Mater
  • and Katashima et al, 2019 Curr Opin Chem Eng.

Two paragraphs, beginning in Line 61 and Line 84, respectively, were rewritten to introduce these new references.

Reviewer 3 Report

My major concern is that the conclusions of this manuscript are mainly based on the representative images. How would the author guarantee other parts of the modified silk demonstrate a similar observation?

Besides, the following concerns need to be addressed before a further evaluation:

Lines 79-80, why does the author mentioned the challenge for the functionalization of PLA and PLG?

It is better to introduce current status of the four strategies for the silk protein functionalization in the section of Introduction;

Table 1, the experiment design for the control group is required;

It looks like the quantity of silk fibers in Figure 1-1 is greater than those of Figures 1-2 and 1-3;

The bright field optical micrographs for Figure 2 shall be presented;

Line 154, full name of SFS shall be presented;

Scale bar for Figure 1 is required;

Table 1, the data is lack of standard deviation;

Lines 188, “the previous study……”, citation(s) is (are) required;

Section 2.2, some description regarding experimental procedure shall be moved to “Materials and Methods” part;

Figure 7, the unit for the x axis should be listed.

Author Response

Reviewer #3

We would like to thank Reviewer #3 for his/her constructive comments that we have dealt with as follows.

  • How would the author guarantee other parts of the modified silk demonstrate a similar observation?.

In those cases in which the results are obtained from the micrographs, at least two images were taken from different regions of the fiber and at least three measurements were taken from each micrograph. This information is now included in the Figure Captions. In addition, other quantitative procedures (absorption and enzymatic assay) were used in some cases to get quantitative data. Those micrographs not used to get quantitative data are labelled as “Illustrative micrographs” in the revised version.

  • Lines 79-80. The challenge of functionalizing PLA and PLG.

A new sentence is added in Line 80 to indicate that the difficulty of functionalizing PLA and PLG is related with the necessity of using the hydroxyl groups present in these artificial polymers, and this chemical reaction does not proceed under physiological conditions.

  • Introduction of the functionalization strategies in the Introduction.

The range of functionalization strategies is expanded in the revised version with the inclusion of 5 new references in the Introduction, as suggested by Reviewer #2.

  • Experimental design in Table 1.

The results in Table 1 were obtained from the dissolution of two yarns, one produced with Protocol A and the other with Protocol B. The number corresponding to each condition corresponds to the value yielded by the spectrophotometer. This detail is indicated in the caption of Table 1 (Ln 194).

  • Quantity of silk in Figures 1.1, 1.2 and 1.3.

These figures are illustrative of the process, as it is now indicated in the Figure caption, and were not used to obtain any quantitative information of the samples. Each micrograph corresponds to a yarn of fibers (a detail now included in the Figure caption), but they do not need to contain the same amount of fibers. Normalization proceeded by measuring the absorption of 40 mg of each yarn, as indicated in the caption of Table 1 (see comment above).

  • Bright fiels images of Figure 2.

No systematic brigth field images were taken along fluorescence micrographs, except in those cases in which no significant fluorescence was obtained. This is especially evident in Figure 4, in which no image of the control fibers is obtained from the fluorescence microscope. This issue is stressed in the Figure caption 4 of the revised version

  • Full name of SFS.

This point was also mentioned by Reviewer #1. The acronym “SFS” is now explained in the Abstract and in the Introduction (Line 99).

  • Scale bar in Figure 1.

A 5 mm scale bar in included in the new Figure 1 included in the revised version.

  • Standard deviation in Table 1.

Since the results included in Table 1 correspond to the readout of the spectrophotometer in each case, no standard deviation is included in this Table. This point is explained in the caption of the Table in the revised version.

  • Citation in Line 188.

We apologize for the initial statement of this sentence, since it was probably confusing. The original sentence:  “The previous study also establishes that the functionalization may proceed…” is now rewritten as “The results above establish that the …”.

  • Move part of the description in Section 2.2. to “Materials and Methods”.

Following Reviewer #3’s suggestion, a new paragraph is added to the “Materials and Methods” section to describe the experimental details of the fluorescence measurements, which was originally explained in Section 2.2.

  • Units in the X-axis of Figure 7.

Figure 7 shows the true stress-true strain curves of representative tests on regenerated silk fibers. In contrast to the plots in which engineering stress and engineering strain are represented, that tend to express strain as a percentage (strain (%)), it is conventional not to include dimensions in the true strain axis, since it corresponds to  the variation of length with respect to the instataneous length of the sample. This question is clarified in the Figure caption 7 of the revised version.

Round 2

Reviewer 1 Report

The authors have addressed all the issues well.

Author Response

I have amended the manuscript following the indications of the Reviewer as follows:

1) The following sentence was added to Figure caption 7:
"(d=dL/L, where  is true strain and L the instantaneous length of the sample)."

A new reference with the details of the mechanical testing procedure is also included: Pérez-Rigueiro et al. J. Appl. Polym. Sci. 1998

2) Amended.

3) Amended

4) Amended

We would like to thank once again the Reviewers for their constructive comments and hope that the manuscript is acceptable now for publication in Molecules.

José Pérez-Rigueiro